# Nicking mechanism underlying the DNA phosphorothioate-sensing antiphage defense by SspE

Haiyan Gao [1,2,6], Xinqi Gong [3,6], Jinchuan Zhou[1], Yubing Zhang[1,2], Jinsong Duan[4], Yue Wei[1,5], Liuqing Chen[2], Zixin Deng [1], Jiawei Wang [4], Shi Chen [1,5] ✉, Geng Wu [2] ✉ & Lianrong Wang [1] ✉

DNA phosphorothioate (PT) modification, with a nonbridging phosphate oxygen substituted by sulfur, represents a widespread epigenetic marker in prokaryotes and provides protection against genetic parasites. In the PT-based defense system Ssp, SspABCD confers a single-stranded PT modification of host DNA in the 5′-$C_{PS}$CA-3′ motif and SspE impedes phage propagation. SspE relies on PT modification in host DNA to exert antiphage activity. Here, structural and biochemical analyses reveal that SspE is preferentially recruited to PT sites mediated by the joint action of its N-terminal domain (NTD) hydrophobic cavity and C-terminal domain (CTD) DNA binding region. PT recognition enlarges the GTP-binding pocket, thereby increasing GTP hydrolysis activity, which subsequently triggers a conformational switch of SspE from a closed to an open state. The closed-to-open transition promotes the dissociation of SspE from self PT-DNA and turns on the DNA nicking nuclease activity of CTD, enabling SspE to accomplish self-nonself discrimination and limit phage predation, even when only a small fraction of modifiable consensus sequences is PT-protected in a bacterial genome.

The unceasing arms race between parasitic viruses and their hosts has driven the emergence of a wide range of antiviral defense machineries and strategies. As a new member of the prokaryotic innate immunity system, the DNA phosphorothioate (PT) modification-based Dnd system is believed to provide protection against foreign DNA invasion in a manner analogous to that of methylation-based restriction-modification barriers[1–3]. DndABCDE acts as the modification component to catalyze the sequence-specific oxygen-sulfur swap in the DNA sugar-phosphate backbone to generate double-stranded PT modifications[4,5].

PT is employed as the recognition tag by the restriction cognate DndFGH or PbeABCD to discriminate and destroy non-PT-modified invasive foreign DNA[2,6]. In response to the loss of DNA PT modification, dndBCDE-deficient *Salmonella enterica* mutant cells undergo self-restriction by unrestrained DndFGH, and the resultant DNA damage triggers the cellular SOS response[3,7].

Knowledge about PT diversity was recently expanded by our identification of Ssp systems, which display genetic organization, enzymatic activities and phenotypic behaviors that are different from

[1]Department of Gastroenterology, TaiKang Center for Life and Medical Sciences, Zhongnan Hospital of Wuhan University, School of Pharmaceutical Sciences, Wuhan University, Wuhan 430071, China. [2]State Key Laboratory of Microbial Metabolism, School of Life Sciences & Biotechnology, The Joint International Research Laboratory of Metabolic & Developmental Sciences, Shanghai Jiao Tong University, Shanghai 200240, China. [3]Mathematics Intelligence Application Lab, Institute for Mathematical Sciences, Renmin University of China, Beijing 100872, China. [4]State Key Laboratory of Membrane Biology, Beijing Advanced Innovation Center for Structural Biology, School of Life Sciences, Tsinghua University, Beijing 100084, China. [5]Department of Burn and Plastic Surgery, Shenzhen Institute of Translational Medicine, Health Science Center, the First Affiliated Hospital of Shenzhen University, Shenzhen Second People's Hospital, Shenzhen 518035, China. [6]These authors contributed equally: Haiyan Gao, Xinqi Gong. ✉e-mail: shichen@whu.edu.cn; geng.wu@sjtu.edu.cn; lianrong@whu.edu.cn

those of Dnd systems[8,9]. In contrast to the double-stranded PT modifications governed by DndABCDE, i.e., 4-bp 5′-$G_{PS}$AAC-3′/5′-$G_{PS}$TTC-3′ (PS, phosphorus-sulfur linkage) consensus sequences in *S. enterica* serovar Cerro 87 and *Escherichia coli* B7A and 5′-$G_{PS}$ATC-3′/5′-$G_{PS}$ATC-3′ in *Hahella chejuensis* KCTC2396, PT modification in SspABCD-expressing *Vibrio cyclitrophicus* FF75 occurs as a single-stranded modification at the 3-bp 5′-$C_{PS}$CA-3′ motif; no PT modification was detected in the complementary 5′-TGG-3′[10,11]. This single-stranded sulfur incorporation is predicted to be attributed to the nicking activity of SspB, which has not yet been detected for any one of the Dnd proteins[8]. Notably, SspABCD coupled with SspE confers resistance to a wide array of phages through unusual mechanisms: (1) SspE is a dual-function protein that exerts both N-terminal NTPase and C-terminal DNA nicking nuclease activities to defend against phage infection, and (2) NTPase activity, including GTPase, CTPase, and UTPase activity, is stimulated exclusively by 5′-$C_{PS}$CA-3′-containing DNA in vitro[8]. These features render SspABCD-SspE an unusual PT-DNA-sensing antiphage mechanism, differing from PT-based Dnd or methylation-based

restriction-modification (R-M) defense. Moreover, genomic PT mapping revealed that only 14% of the 160,541 5′-CCA-3′ sites across the *V. cyclitrophicus* FF75 genome are PT-modified and that PT distribution is heterologous in a population of DNA molecules[10,12]. These unusual features raise the question of the fundamental importance of how SspE utilizes epigenetic PT modification to accomplish self-nonself discrimination and to coordinate its NTPase and DNA nicking activities to fight against phages.

In this work, we locate a hydrophobic patch on the surface of the N-terminal domain (NTD) of SspE that is responsible for the specific recognition of the 5′-$C_{PS}$CA-3′ PT modification. Strikingly, through the joint action of this PT-sensing patch and a DNA-binding domain in the CTD, SspE exhibits a PT-binding preference, which prevents the non-PT-protected regions from being targeted by SspE. Upon PT recognition, the NTP binding pocket adjacent to the PT-sensing patch becomes enlarged, which expedites NTP access and thereby enhances NTP hydrolysis activity. By combining normal mode analysis and fluorescence resonance energy transfer (FRET) measurements in vitro,

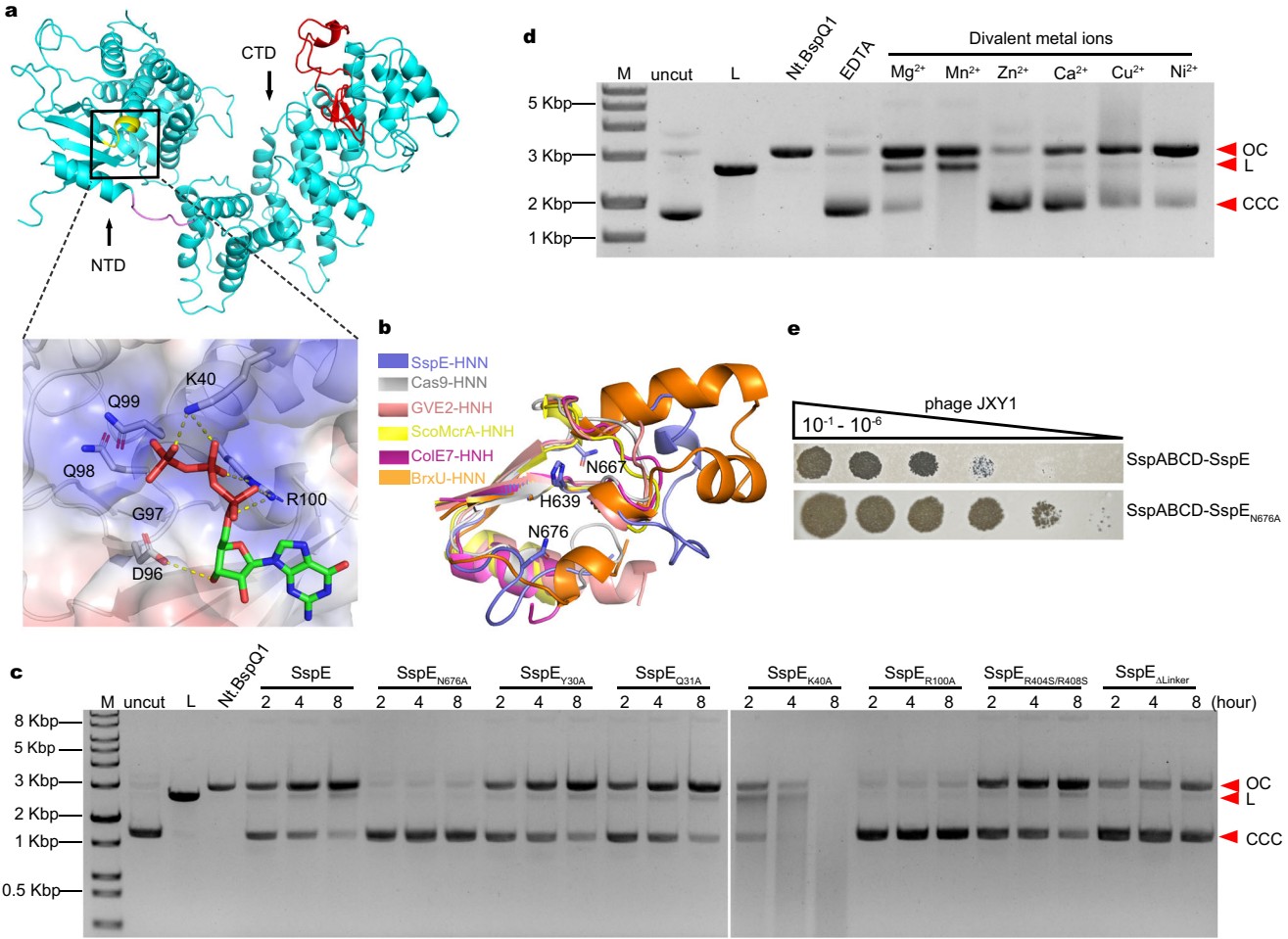

**Fig. 1 | Crystal structure of the full-length SspE from S. yokosukanensis DSM 40224. a** Ribbon diagram of full-length SspE with the NTD and CTD shown in cyan and the 7-aa linker shown in magenta. Yellow and red indicate the $D^{96}GQQR^{100}$ motif in the NTD and HNH motif in the CTD, respectively. The lowest-energy docking structure of the NTD with GTP is shown as a black box. Hydrogen bonding interactions (distance cutoff: 3.6 Å) are displayed as yellow dots. **b** Superposition of the HNH motifs from SspE (PDB: 7DRS, slate), Cas9 (PDB: 6O56, gray), GVE2 (PDB: 5H0O, salmon), ScoMcrA (PDB: 5ZMM, yellow), ColE7 (PDB: 1M08, magenta) and BrxU (PDB: 7P9K). Catalytic residues are positioned as indicated. **c** Effect of mutations of the indicated residues on the in vitro DNA nicking activity of SspE. Supercoiled pUC19 was used as a substrate in DNA nicking assays. The impaired

nicking activity of $SspE_{R100A}$ was used as a control. The nicked DNA products were analyzed on two 1% agarose gels. Two gels run under identical conditions were cropped and fused together. EcoRI-linearized pUC19 and Nt. BspQI-nicked pUC19 was used as a reference. CCC, covalently closed circular DNA; OC, open circular DNA; L, linear DNA. All results are representative of three independent experiments. **d** Tests of the divalent cation requirements for the nicking activity of SspE toward pUC19. Data are representative of two independent experiments. **e** Plaque assays of *S. lividans* HXY6 expressing SspABCD-SspE and SspABCD-SspE$_{N676A}$ using 2 μL of each serial tenfold dilution ($10^{-1}$–$10^{-6}$) of phage JXY1 at 28 °C. Source data are provided as a Source Data file.

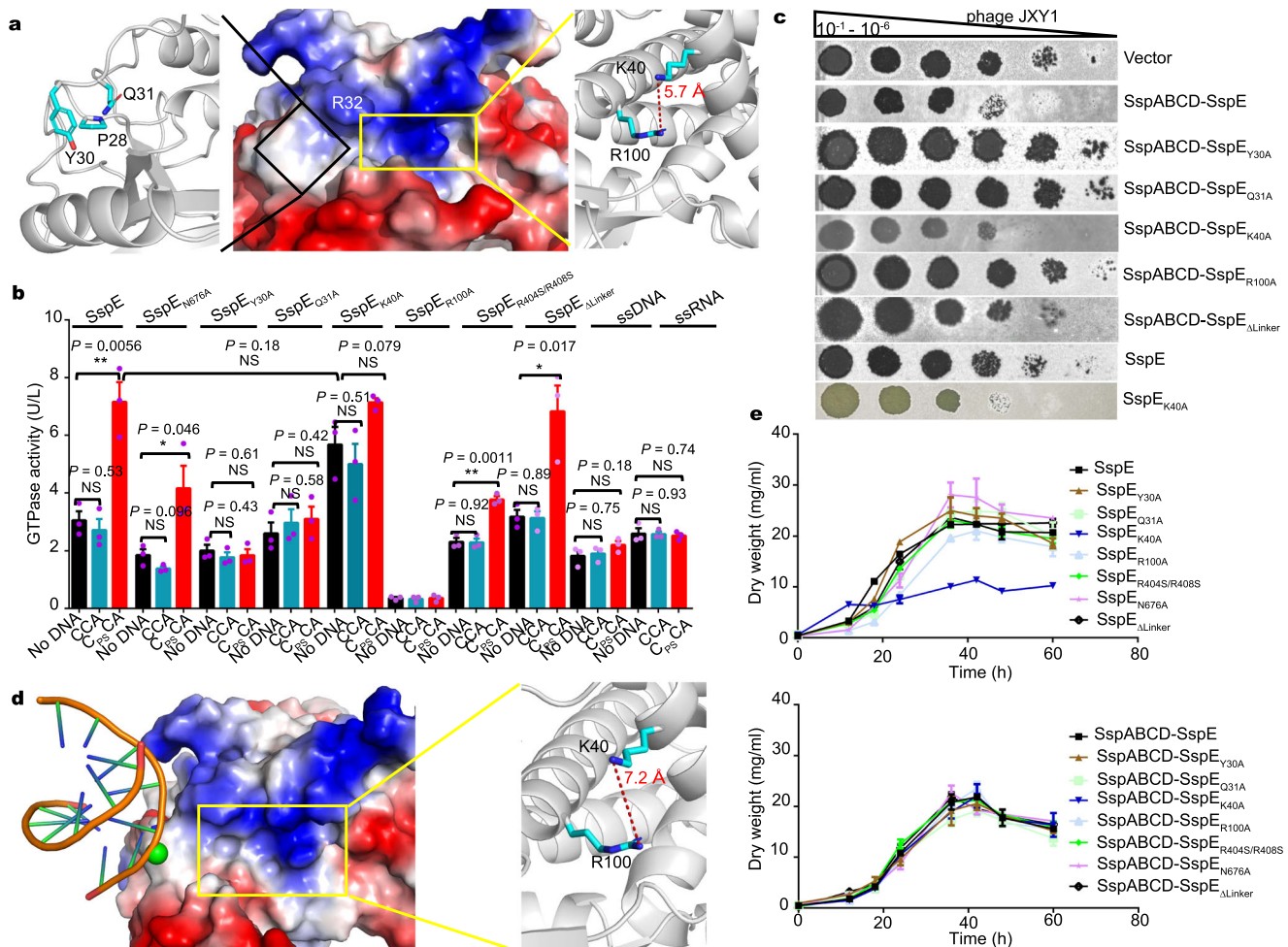

**Fig. 2 | Structural, biochemical, and antiviral analyses of SspE and its variants.**
**a** The distribution of the PT-sensing cavity and $D^{96}GQQR^{100}$-containing NTP binding pocket are shown as black and yellow boxes, respectively, in the NTD of SspE. Key residues are represented as sticks. The magenta dotted line represents the distance of side chains between residues K40 and R100 in the absence of PT-DNA.
**b** Assessment of the GTPase activity of SspE and its mutants in the presence of 40-bp 5'-CCA-3'- or 5'-$C_{PS}$CA-3'-containing DNA fragments. The responsiveness of SspE GTPase activity to 5'-CCA-3'-containing single-stranded DNA or RNA was also measured. The mean ± SEM values of three independent experiments are shown. Statistical significance was calculated by unpaired two-sided Student $t$ tests. **$P < 0.01$. *$P < 0.05$. NS, not significant. **c** Plaque assays were carried out to compare the antiphage activities of SspE and variants in *S. lividans* HXY6 using 2 μL of each serial tenfold dilution ($10^{-1}–10^{-6}$) of phage JXY1 at 28 °C. The data shown here are representative images of three independent experiments. **d** Docking of double-stranded DNA (5'-GCGTCCA-3'/5'-TGGACGC-3' derived from PDB entry 2XHI) into

the hydrophobic patch on the $SspE_{NTD}$ surface. The surface of $SspE_{NTD}$ is colored according to the electrostatic potential. Red and blue indicate negative and positive potentials, respectively. The sulfur atom on the DNA skeleton is colored green. The magenta dotted line represents the distance of side chains between residues K40 and R100 in the docking structure of $SspE_{NTD}$ with 5'-$C_{PS}$CA-3'-containing DNA. **e** Growth curve analysis of *S. lividans* HXY6 cells expressing wild-type SspE or its mutants grown in TSBY medium (3% tryptone soy broth medium, 0.5% yeast extract, and 10.3% sucrose) at 28 °C for 72 h (upper panel). A 1-mL aliquot was collected at the indicated time points and centrifuged for 30 min at 16,000 × g. The supernatant was discarded, and the pellet was dried at 70 °C for 48 h and weighed. $SspE_{K40A}$ alone attenuated the growth of PT-lacking *S. lividans* HXY6. Effects of SspE and its mutants on cell growth in the presence of SspABCD (lower panel). Values represent the mean of triplicate experiments for each time point tested. Error bars represent the SD from $n = 3$ replicates. Source data are provided as a Source Data file.

we demonstrate that SspE undergoes a conformational switch between a 'closed' and an 'open' state induced by PT-stimulated GTP hydrolysis. The closed-to-open transition simultaneously promotes the dissociation of SspE from PT-DNA and turns on the nickase activity of CTD. In summary, this study reveals an exquisite molecular mode of action of SspE as an epigenetic PT-sensing defense mechanism against phage infection.

## Results

### Determination of the structure of full-length SspE

In our previous work, we identified that SspE relies on a 5'-$C_{PS}$CA-3' PT modification in host DNA to defend itself against phage infection, and we determined the crystal structure of full-length SspE from *Streptomyces yokosukanensis* DSM 40224[8]. However, the side chains

of the C-terminal half, including residues 327-771, could not be resolved due to the mediocre resolution of the X-ray diffraction[8]. To explore the molecular mechanism of the unusual PT-sensing phage resistance of SspE, we successfully determined the crystal structure of the CTD of SspE (hereafter referred to as $SspE_{CTD}$) from *Streptomyces scabiei* DSM 41658 to 2.7 Å resolution (PDB accession number 7DRI) through the single-wavelength anomalous dispersion (SAD) method by using a selenomethionine (SeMet) derivative (Supplementary Table 1). The crystal structure had four molecules in the a-symmetric unit of $SspE_{CTD}$ (Supplementary Fig. 1a). Each molecule exhibited an α-helical fold with only a handful of β-sheets (Supplementary Fig. 1b).

This $SspE_{CTD}$ structure was subsequently used as a molecular replacement search model to determine the positions of the side

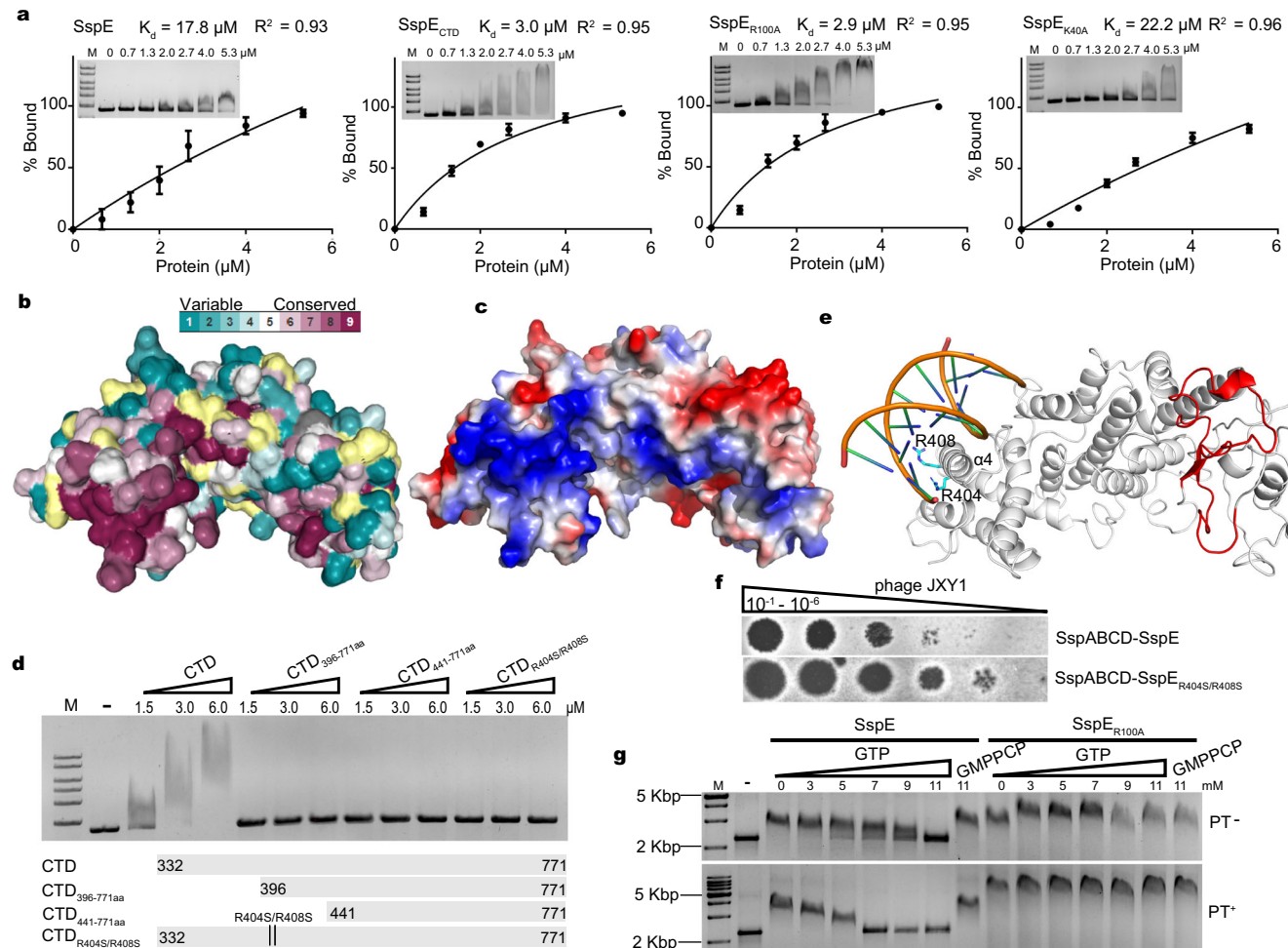

**Fig. 3 | Structural basis for the DNA binding of SspE$_{CTD}$. a** DNA binding assessment by EMSA. Different concentrations of SspE and its mutants were incubated with 200 ng of linearized pUC19 DNA in 20 mM Tris·HCl (pH 8.0), 100 mM NaCl, and 5% glycerol for 15 min at 30 °C before loading onto a 1% agarose gel. The intensity of the shifted DNA band is expressed as the percentage bound. A non-linear regression analysis was applied to the datasets using GraphPad Prism software (version 6) to obtain the best-fit curve. Values represent the mean of three biological replicates, and error bars represent standard deviations. **b** Surface representation of the CTD of SspE colored according to sequence conservation. **c** Electrostatic surface representation of the CTD of SspE with negative potential shown in red and positive potential shown in blue. **d** EMSA analysis of the binding of SspE$_{CTD}$ and its mutants, shown schematically, with increasing concentrations of EcoRI-linearized pUC19 DNA. All results are representative of three independent experiments. **e** Model for the binding of SspE$_{CTD}$ to a 10-bp DNA fragment (PDB: 5J3G). The conserved positively charged residues, R404 and R408, mutated in this study are represented as sticks. The HNH motif is shown in red. **f** Plaque assays of *S. lividans* HXY6 expressing SspABCD-SspE and SspABCD-SspE$_{R404S/R408S}$ using 2 μL of each serial tenfold dilution ($10^{-1}$–$10^{-6}$) of phage JXY1 at 28 °C. **g** EMSAs of SspE and SspE$_{R100A}$ binding to PT-modified or unmodified pUC19 DNA substrates in the presence of GTP/GMPPCP. The results are representative of two independent experiments. Source data are provided as a Source Data file.

chains of the CTD residues of full-length SspE from *S. yokosukanensis* DSM 40224. The structure was refined to an $R_{work}/R_{free}$ value of 20.1%/27.6% at 3.4 Å resolution using the REFMAC program[13] (Supplementary Table 2). The full-length SspE structure revealed a smaller NTD (residues 1 to 324) and a larger CTD (residues 332 to 771), which were connected by a short linker of only seven residues T$^{325}$TGQPLT$^{331}$ (Fig. 1a). The NTD of SspE exhibited a DUF262-fold with the strongly conserved D$^{96}$GQQR$^{100}$ signature motif, which was essential for GTP hydrolysis and was positioned at the junction of the α3 helix and the β4 strand (Fig. 1a). Fig. 1a displays the lowest-energy (−7.003 Kcal/Mol) docking structure of the GTP-SspE$_{NTD}$ complex, in which GTP is capable of forming hydrogen bonds to the side chains of D96 and R100 within the D$^{96}$GQQR$^{100}$ motif as well as K40. This structure provides an explanation for our previous observation that the single replacement of R100 with an alanine abolishes the GTP hydrolysis activity of SspE$_{R100A}$[8].

The CTD of SspE contains a DUF1524 domain that belongs to the His-Asn-His (HNH, the last histidine of which can be replaced by an asparagine, as observed in SspE) nuclease superfamily and indeed exerts DNA nickase activity in vitro[8]. In contrast to the typical HNH motif in Cas9[14], GVE2[15], ScoMcrA[16], ColE7[17], or BrxU[18] that consists of two antiparallel β-sheets and an α-helix, the HNH motif in the CTD of SspE from *S. yokosukanensis* DSM 40224 was composed of two antiparallel β-sheets and a loop, suggesting comparatively higher flexibility (Fig. 1b). However, the active sites of the HNH motif in SspE still adopt a characteristic ββα-metal topology, in which the first histidine typically acts as the general base in the DNA cleavage reaction, asparagine is involved in structural stabilization and the third residue, which can be either histidine or asparagine, is responsible for metal binding[19,20]. Indeed, single-point substitution of N676 in the HNH motif by an alanine or the elimination of divalent metal cations, i.e., Mg$^{2+}$, markedly impaired the DNA nicking activity of SspE in vitro (Fig. 1c and d). This finding was in line with the results showing that SspABCD-SspE$_{N676A}$ lost resistance against phage JXY1 infection in *Streptomyces lividans* HXY6, which lacked endogenous PT modification genes (Fig. 1e).

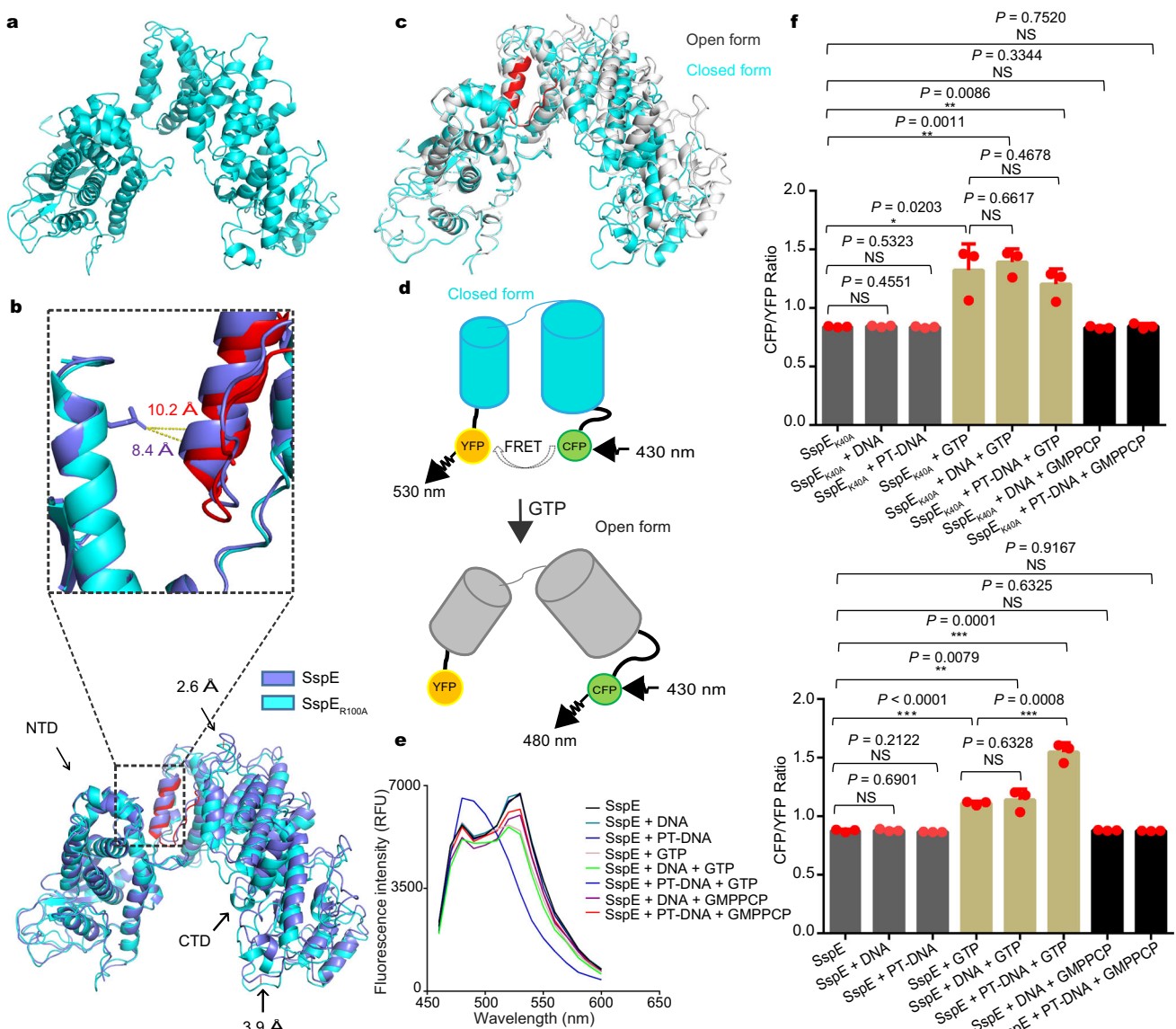

**Fig. 4 | The structure of SspE$_{R100A}$ and conformational changes in SspE. a** The structure of SspE$_{R100A}$ was determined, and one of the four monomers in one asymmetric unit is displayed. **b** Structural superimposition of the wild-type SspE structure (slate) with SspE$_{R100A}$ (cyan) to show that CTD exhibits *en bloc* movement toward the NTD. The DNA binding region of SspE$_{R100A}$ is highlighted in red. The distance between the R404- and R408-containing helix and the NTD in SspE$_{R100A}$ is greater than that in SspE. **c** The "open" and "closed" conformational states of SspE by normal mode analysis. The DNA binding region of SspE in the closed conformational state is highlighted in red. **d** A schematic representation describing the FRET changes of YFP-SspE-CFP in the closed-to-open transition. In the closed state, FRET occurs from excited CFP to YFP, leading to light emission from YFP at 530 nm. GTP hydrolysis causes a conformational change in SspE, leading to reduced emission from YFP and increased emission from CFP at 480 nm. **e** Spectra of FRET monitoring of YFP-SspE-CFP in vitro in the absence or presence of PT-DNA and/or GTP (left panel). FRET was expressed as the emission ratio of CFP to YFP signals (right panel). Data and error bars represent the mean ± SD from three independent experiments. Statistical significance was calculated by unpaired two-sided Student *t* tests; **$P$ < 0.01, ***$P$ < 0.001, and NS, not significant. **f** FRET analysis of YFP-SspE$_{K40A}$-CFP in vitro. FRET was reported as the emission ratio of CFP to YFP signals. Data and error bars represent the means ± SD from three independent experiments. Statistical significance was calculated by unpaired two-sided Student *t* tests; *$P$ < 0.05, **$P$ < 0.01, and NS, not significant. Source data are provided as a Source Data file.

## SspE functions as a PT reader, rendering SspABCD-SspE a PT-sensing defense machinery

It is striking that the NTPase activity, as exemplified by its GTPase activity, of SspE is stimulated specifically by 5′-C$_{PS}$CA-3′-containing DNA[8], which prompted us to determine the PT-sensing molecular mechanism. Inspired by the knowledge that the sulfur atom in the PT linkage is negatively charged and more hydrophobic than the equivalent oxygen present in native phosphodiester[16,21], we focused on hydrophobic regions likely involved in the association with DNA PT modification. As shown in Fig. 2a, one unique hydrophobic patch that was also lined with positively charged residues, e.g., R32, was located

on the surface of the N-terminal region of the SspE structure. Mutations of residues located in this hydrophobic cavity revealed that the single-point mutations Y30A and Q31A completely abolished the responsiveness of SspE to the PT modification, as manifested by the lack of response of the GTPase activities of SspE$_{Y30A}$ and SspE$_{Q31A}$ to the 5′-C$_{PS}$CA-3′-containing DNA fragment[8] in vitro compared to the PT-stimulated GTPase activity of wild-type SspE (Fig. 2b). While SspE$_{Y30A}$ and SspE$_{Q31A}$ still sustained GTP hydrolysis activities comparable to that of nonstimulated wild-type SspE, SspABCD-SspE$_{Y30A}$ and SspABCD-SspE$_{Q31A}$ of *S. yokosukanensis* DSM 40224 on plasmids pPT586 and pPT585, respectively, no longer provided *S. lividans* HXY6

with resistance to phage JXY1 (Fig. 2c). The results demonstrated that (1) the Y30- and Q31-containing hydrophobic patch played an essential role in mediating the PT-sensing activity of SspE, and (2) PT-stimulated GTP hydrolysis activity, albeit only at a level approximately 2.3-fold higher than the intrinsic GTPase activity, is critical to induce the anti-phage defense of SspE.

When 5′-C$_{PS}$CA-3′ was replaced by a motif such as 5′-G$_{PS}$AAC-3′ or 5′-G$_{PS}$GCC-3′[8] or occurs in a single-stranded DNA or RNA molecule, no stimulatory effects on GTPase activity were detected (Fig. 2b), highlighting the complicated interactions between 5′-C$_{PS}$CA-3′-containing DNA and SspE for sequence-specific PT reading. However, an electrophoretic mobility shift assay (EMSA) showed that the NTD of SspE (SspE$_{NTD}$) did not display considerable binding affinity for the 5′-C$_{PS}$CA-3′-containing DNA (Supplementary Fig. 2), suggesting either a transient interaction or weak binding affinity. Therefore, we performed molecular docking of PT-DNA into the hydrophobic cavity of the SspE$_{NTD}$ structure using the holistic docking pipeline HoDock[22]. Interestingly, the docking simulation results showed that upon complex formation with 5′-C$_{PS}$CA-3′-containing DNA, the entrance to the NTP-binding pocket adjacent to the PT-sensing cavity widened significantly, as manifested by an increased distance between residues K40 and R100 (Fig. 2a and d). This conformational change in SspE$_{NTD}$ was predicted to result from the electrostatic interaction between the negatively charged backbone phosphate of PT-DNA and the positively charged residues in the NTP binding pocket. To test this hypothesis, we substituted the positively charged K40, located near the entrance of the NTP binding pocket, with an alanine and performed GTPase activity assays (Fig. 2b). Indeed, the SspE$_{K40A}$ mutant was no longer responsive to PT-DNA but exhibited GTPase activity ($5.7 \pm 0.6$ U/L) comparable to the PT-stimulated GTPase activity of wild-type SspE ($7.2 \pm 0.7$ U/L), suggesting that the side chain of K40 sterically hindered GTP entry (Fig. 2b). Collectively, these results unraveled the molecular recognition mechanism by which SspE functioned as a PT-stimulated NTPase, in which its hydrophobic and electrostatic interactions with 5′-C$_{PS}$CA-3′-containing DNA stimulated NTP access into the catalytic pocket. Upon interacting with PT-DNA, the entrance of the NTP binding pocket widened, allowing easier access for substrates, i.e., GTP, and consequently led to an increase in NTP hydrolysis activity. SspE$_{K40A}$, exhibiting constitutively enhanced GTPase activity, could therefore be regarded as a mimic of wild-type SspE that had been stimulated by PT-DNA. The constitutive GTPase stimulation of the NTD yielded a PT-independent, constitutively active nicking nuclease in the CTD, as reflected by the observations that SspE$_{K40A}$ exhibited much higher DNA nicking activity than wild-type SspE in vitro (Fig. 1c). Consequently, this mutant exerted a detrimental effect on growth of the *S. lividans* HXY6 host, even in the absence of the SspABCD-mediated 5′-C$_{PS}$CA-3′ PT modification (Fig. 2e). This phenomenon is in sharp contrast to the SspE$_{R100A}$ mutant, which simultaneously lost the GTPase activity of NTD, the DNA nickase activity of CTD, and the resistance against phage infection[8].

## GTP hydrolysis modulates the affinity of SspE for DNA

Although the EMSA results established that SspE$_{NTD}$ had no DNA binding affinity, the full-length SspE displayed an obvious affinity for the EcoRI-linearized pUC19 DNA substrate with a binding dissociation constant ($K_d$) of 17.8 μM, indicating the involvement of the CTD in DNA binding (Fig. 3a and Supplementary Fig. 2). Since DNA generally binds to a positively charged surface on its target protein, we first analyzed the surface conservation of the SspE$_{CTD}$ structure and simultaneously searched for positively charged residues in the highly conserved regions (Fig. 3b, c). Then, the conserved positively charged amino acid residues of *S. yokosukanensis* DSM 40224 SspE were selected for mutation or truncation (Fig. 3d). The resultant SspE$_{CTD}$ variants were next incubated with linearized pUC19 DNA for EMSA analysis, which revealed that the N-terminal end of the CTD was responsible for DNA binding (Fig. 3d). Based on molecular docking analysis, we established

a structural model of SspE$_{CTD}$ in complex with DNA, which showed that the N-terminal region of the CTD bound to DNA through the R404- and R408-containing α4 helix and a loop (Fig. 3e). Indeed, the SspE$_{R404S/R408S}$ mutant lost DNA binding activity in vitro, and the SspABCD-SspE$_{R404S/R408S}$ module in *S. lividans* HXY6 no longer provided resistance to phage JXY1 infection, which demonstrated the necessity of the DNA binding activity for the defensive function of SspE (Fig. 3f).

Notably, the quantitative EMSA results showed that SspE$_{CTD}$ bound more tightly to the DNA substrate ($K_d = 3.0$ μM) than the full-length SspE ($K_d = 17.8$ μM), indicating that the NTD might negatively regulate the DNA binding activity of the CTD (Fig. 3a). To test this hypothesis, we compared the DNA binding affinities of SspE$_{K40A}$ and SspE$_{R100A}$. Remarkably, the GTPase-defective SspE$_{R100A}$ mutant bound to pUC19 DNA as strongly as SspE$_{CTD}$, whereas SspE$_{K40A}$ with enhanced GTPase activity bound to the DNA substrate with a $K_d$ (22.2 μM) comparable to that of full-length SspE (Fig. 3a). Therefore, a tempting speculation was that upon sensing PTs in host DNA, the GTPase activity of NTD was stimulated, subsequently promoting the dissociation of the CTD from PT-modified self-DNA. This speculation was supported by three in vitro observations: (1) as the concentration of GTP increased, SspE was released from the SspE-DNA complex, (2) PT-stimulated GTP hydrolysis led to a faster release of SspE from PT-DNA, and (3) GTPase-inactive SspE$_{R100A}$ remained tightly bound to DNA when exposed to the same GTP conditions (Fig. 3g). In addition, the DNA-binding activity of SspE was not susceptible to β,γ-methylene-guanosine 5′-triphosphate (GMPPCP), a nonhydrolyzable GTP analog, indicating that the affinity of SspE for DNA was modulated in a GTP hydrolysis-dependent manner (Fig. 3g).

## GTP hydrolysis triggers a conformational switch

Considering that the NTD of SspE inhibited the enzymatic activities, i.e., DNA binding and nicking nuclease activities, of the CTD, it raised a possibility of intramolecular interactions between the NTD and CTD. To test this hypothesis, we first deleted G$^{327}$QP$^{329}$ from the 7-aa linker region between the NTD and CTD. Indeed, the resulting SspE$_{\Delta linker}$ maintained responsiveness to the PT modification in a manner similar to wild-type SspE, but the C-terminal nickase activity and protection against phage infection were obviously reduced (Fig. 1c, Fig. 2b, c). We then determined the crystal structure of SspE$_{R100A}$, a mutant with both N-terminal GTPase activity and C-terminal nickase activity deficiency, to a resolution of 3.48 Å (Fig. 4a and Supplementary Table 4). When the structures of the NTDs of SspE$_{R100A}$ and wild-type SspE were superimposed, the CTD of SspE$_{R100A}$ was observed to have an *en bloc* movement of 2.6-3.9 Å toward the NTD domain (Fig. 4b). As a result of this conformational change, the R404- and R408-containing helix moved away from the NTD, yielding a larger space for DNA (Fig. 4b). This result was consistent with the observation that SspE$_{R100A}$ exhibited a DNA binding affinity similar to that of SspE$_{CTD}$ and stronger than that of wild-type SspE (Fig. 3a).

Given the conformational difference between SspE$_{R100A}$ and wild-type SspE, we next wondered whether wild-type SspE underwent a conformational change while functioning. SspE displayed an extended dumbbell-like shape, with a short loop linking its two globular NTD and CTD domains. The NTD and CTD packed loosely against each other, with few contacts between them. When the flexibility of SspE was analyzed by normal mode analysis, a computational approach for the analysis of collective motions in biomolecules, an intrinsic vibrating motion between the two domains of SspE was found, resulting in its conformation alternating between an "open" and a "closed" state (Fig. 4c). Strikingly, the structure of SspE$_{R100A}$ resembled that of the predicted "closed" form of wild-type SspE with a root mean square deviation (RMSD) value of 1.724 Å for 622 aligned Cα atoms. An interesting possibility is that SspE may exert PT-sensing defense against phage infection by alternating between the "open" state when PT-DNA is

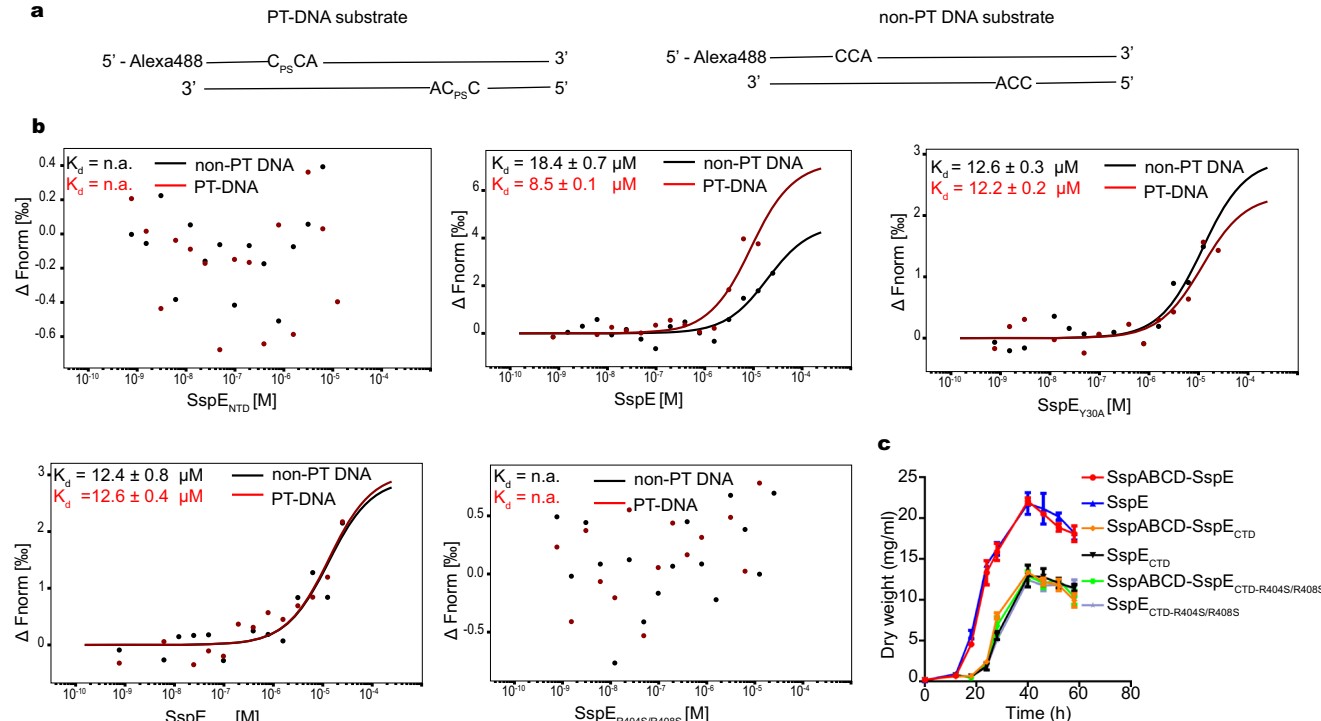

**Fig. 5 | PT modulates the self-nonself discrimination of SspE. a** The 5′-$C_{PS}$CA-3′ and 5′-CCA-3′ containing DNA substrates are shown schematically. **b** MST assay measuring the effect of PT modification at 5′-$C_{PS}$CA-3′ on binding interactions between SspE and variants with DNA. SspE and variants were titrated in a fixed concentration (12.5 nM) of DNA substrates. $SspE_{NTD}$ and $SspE_{R404S/R408S}$ did not produce binding curves regardless of the presence of PT-modified or unmodified DNA substrates. The change in normalized fluorescence (ΔFnorm in ‰) is plotted against the concentration of SspE or its variants. Data are shown as the mean ± SD of three independent experiments. **c** Without the self-recognition mediated by the joint action of the PT-reading hydrophobic cavity and R404- and R408-containing DNA binding affinity, the nickase-active $SspE_{CTD}$ and $SspE_{CTD-R404S/R408S}$ were detrimental to the growth of *S. lividans* even in the presence of SspABCD. SspABCD-SspE or full-length SspE alone exhibited no toxicity to cell growth. Experiments were repeated three times, and each point represents the mean ± SD. Source data are provided as a Source Data file.

present and the "closed" state when PT-DNA is absent. This hypothesis is verified by two observations: (1) SspE alone results in no deleterious autoimmunity effects on PT-lacking cells in vivo, indicating the "closed" conformation of SspE, but (2) nickase activity is turned on once GTPase activity is stimulated, as manifested by the toxicity of $SspE_{K40A}$ toward cell growth (Fig. 2e). We therefore attempted to optically detect the conformational change by flanking the entire SspE protein with N-terminal YFP and C-terminal CFP and monitoring fluorescence resonance energy transfer (FRET) between the two fluorophores (Fig. 4d). Upon GTP hydrolysis, the YFP-SspE-CFP construct exhibited an apparent FRET change, as manifested by the increased ratio of emissions at 480 nm (CFP) to 530 nm (YFP). The increase in the CFP/YFP ratio was greater upon further addition of PT-modified pUC19 DNA (Fig. 4e). In terms of $SspE_{K40A}$ that exhibited enhanced GTPase activity in a PT-independent manner, we observed that the addition of GTP increased the FRET ratio to the same extent as GTP plus PT-DNA (Fig. 4f). This finding confirmed that GTP hydrolysis led to a longer distance between the NTD and CTD, in agreement with the conformational transition from the 'closed' to the 'open' form (Fig. 4e).

## SspE exhibits a PT-binding preference

After determining how PTs turned on the nickase activity of SspE, an unanswered question remained about how the bacterial DNA regions possessing non-PT-modified 5′-CCA-3′ motifs are protected from SspE because only ~14% of genome-wide 5′-CCA-3′ sites were PT modified and PT patterns were heterologous among a population of DNA molecules[12,23]. To address this question, we compared the interactions of SspE with PT-DNA and non-PT DNA. First, we generated a 2-kb DNA fragment harboring two 5′-$C_{PS}$CA-3′ sites on the 5′ and 3′ termini

through PCR amplification (Fig. 5a). Using microscale thermophoresis (MST), a highly sensitive technique to quantify biomolecular interactions, a $K_d$ of 18.4 μM was observed for the interaction of wild-type SspE with the non-PT DNA substrate (Fig. 5b). The MST results were in good agreement with the aforementioned quantitative EMSA data, which revealed a dissociation constant of 17.8 μM for the binding of SspE with unmodified pUC19 DNA. Interestingly, SspE bound to the 5′-$C_{PS}$CA-3′-containing DNA substrate with a $K_d$ = 8.5 μM, representing a 2.2-fold stronger SspE-binding affinity than that of its non-PT counterpart (Fig. 5b). However, this preferential affinity to PT-DNA was not detected for the $SspE_{Y30A}$ and $SspE_{Q31A}$ mutants, which showed abrogated PT responsiveness (Fig. 5b). Moreover, without the R404- and R408-containing DNA binding region, $SspE_{R404S/R408S}$ produced no obvious binding curve with DNA substrates even in the presence of the PT-reading hydrophobic patch (Fig. 5b). Thus, we identified that the PT-binding preference was attributed to the sum of PT responsiveness of NTD and the DNA binding affinity of CTD, and the former was responsible for directing SspE toward the 5′-$C_{PS}$CA-3′-containing region. This result supports the model that the PT-binding preference prevents non-PT-modified 5′-CCA-3′ sites in self-DNA from being targeted by SspE to avoid autoimmunity.

We assessed the toxicity of $SspE_{CTD}$ and $SspE_{K40A}$, both of which have constitutively active nuclease activities, to further examine this model. Upon the loss of PT responsiveness in NTD, $SspE_{CTD}$ was incapable of accomplishing self-recognition and exerted a detrimental effect on bacterial growth, even in the presence of the 5′-$C_{PS}$CA-3′ modification in host DNA (Fig. 5c). In sharp contrast, although $SspE_{K40A}$ also impaired the growth of PT-lacking HXY6 like $SspE_{CTD}$, it no longer led to deleterious autoimmune effects on HXY6 cells when

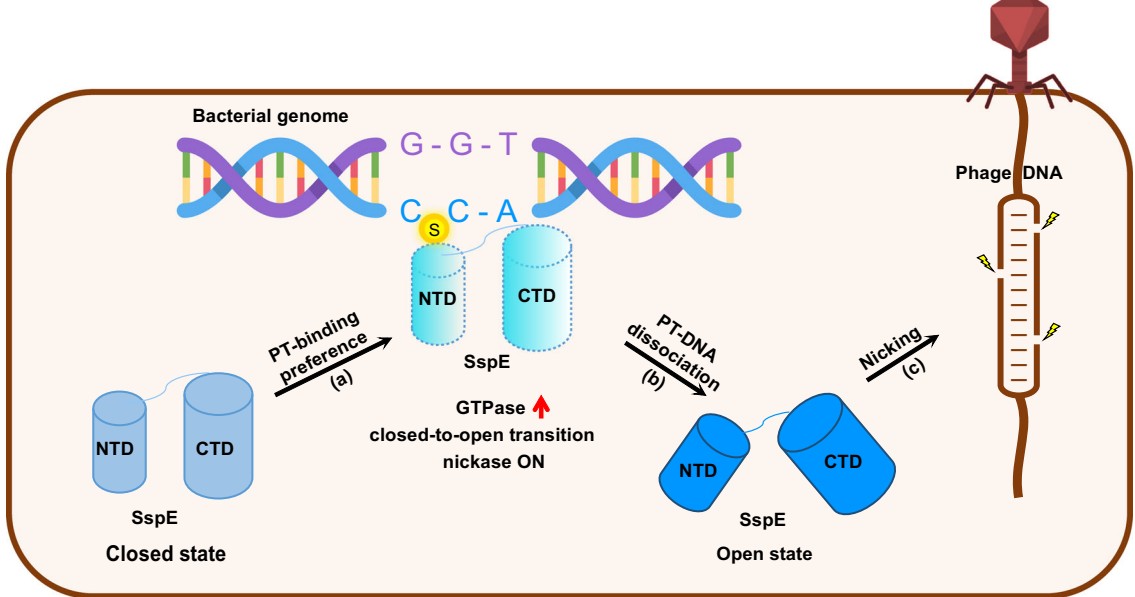

**Fig. 6 | A schematic model of the coupled recognition-activating-nicking mechanism and defense against phage infection. a, b** Upon PT sensing, SspE is recruited to preferentially bind PT sites, where its GTP hydrolysis is stimulated followed by the closed-to-open transition and dissociation from the PT-modified bacterial genome. **c** The closed-to-open transition turns on the nickase activity of CTD to introduce nicking damage to non-PT invasive DNA.

co-expressed with SspABCD. Moreover, the SspABCD-SspE$_{K40A}$ module still provided protection against phage infection at the same level as wild-type SspABCD-SspE (Fig. 2c), highlighting the essential role of the hydrophobic patch-mediated PT sensing in preventing autoimmunity. The phenomenon is reminiscent of our recent observations that despite molecule-to-molecule heterogeneity, PT modifications are maintained at a constant density in a genome with spacing distances between two neighboring 5′-C$_{PS}$CA-3′ sites predominantly occurring in a range of less than 10 kb[12,23]. This result supports the key role of PT density in a range of DNA to recruit and restrain SspE such that the non-PT-modified regions in self-DNA are protected from SspE attack.

As PT internucleotide linkage displays enhanced nuclease resistance[24], SspE is generally presumed to recognize the same consensus sequences, i.e., 5′-CCA-3′, with the modification component SspABCD, but it is incapable of nicking 5′-CCA-3′ once it is PT-protected, similar to REases in methylation-based R-M systems. Surprisingly, PT-modified plasmid DNA isolated from SspABCD-expressing cells were still susceptible to the nuclease activities of SspE and SspE$_{K40A}$ in vitro even in the absence of GTP (Supplementary Fig. 3). The discrepancy from the in vivo observations implied that multiple elements are involved in resisting the toxicity of SspE, such as nuclease-resistant PT bonds, DNA recognition sequences longer than 5′-CCA-3′ and DNA secondary structures. Collectively, we have described an unusual epigenetic PT modification-modulated antiphage model in which nuclease-inactive SspE is directed to PT modification sites to stimulate the GTPase activity of the NTD, the closed-to-open conformational transition, dissociation from self-DNA and activation of nickase activity of the CTD, enabling the resulting nuclease-active SspE to destroy invading phage DNA while avoiding self-targeting (Fig. 6).

## Discussion

The DNA PT modification has evolved versatile functions in cell defense, epigenetic regulation, and the maintenance of cellular redox homeostasis[1,11,25]. The present study provides structural and biochemical evidence that the increased hydrophobic feature enables DNA PT modification to act as a switch to modulate the SspE defensive function, which extends our understanding of the role of this epigenetic DNA modification. It should be mentioned that in our study, we observed that the SspABCD-SspE module provided negligible protection against

plasmid transformation in *E. coli*, suggesting that SspABCD-SspE might be a defense system specifically targeting invading phages (Supplementary Fig. 4).

In the present study, we first elucidated the molecular mode of action of SspE as a PT "reader" to specifically sense epigenetic 5′-C$_{PS}$CA-3′ modification and trigger sequential enzymatic activities. In comparison to the phosphodiester counterpart, the PT diester has a greater hydrophobic character, which renders it specifically recognizable by the surface cavity on SspE. Based on the assessment of GTPase activity and the antiphage defensive function of SspE and its variants, we first confirmed that SspE is capable of sensing 5′-C$_{PS}$CA-3′ through the Y30- and Q31-containing hydrophobic patch in the NTD. This phenomenon is reminiscent of the complex interactions between 5′-G$_{PS}$GCC-3′-containing DNA and ScoMcrA, an endonuclease capable of binding and cleaving DNA ~23 bp away from the PT linkage within 5′-G$_{PS}$GCC-3′[16,26]. Upon recognition of 5′-G$_{PS}$GCC-3′ via a sulfur-binding domain (SBD), ScoMcrA undergoes a substantial conformational change to be better accommodated into the major groove of DNA. Moreover, an 80° outward rotation of the sulfur atom occurs so that the 5′-G$_{PS}$GCC-3′ consensus fits into the SBD cavity of ScoMcrA[16,26]. It is thus conceivable that more protein–DNA interactions should be expected to ensure that SspE "reads" PT modification in a 5′-C$_{PS}$CA-3′-specific manner. Second, SspE functions as a PT-dependent regulator to control its own enzymatic activities by inducing conformational changes. Upon recognition of the 5′-C$_{PS}$CA-3′-containing DNA, the GTP binding pocket that resides adjacent to the PT-sensing cavity becomes enlarged, leading to enhanced GTP hydrolysis activity. The PT-stimulated GTPase activity triggers a closed-to-open structural change in SspE, leading to the dissociation of SspE from self-DNA and activation of the C-terminal nuclease activity. Notably, the 5′-C$_{PS}$CA-3′ modification induces only a moderate increase in GTPase activity, which probably contributes to a rapid ON/OFF switching of the SspE defense to address the constant threat of phage predation under physiological conditions. While biochemically interesting, these results left unanswered the question of whether SspE undergoes an opposite open-to-closed conformational change once its GTPase activity is no longer sufficient to maintain the open state of SspE. Afterall, the constitutively active nickase activity of SspE in bacterial cells will increase the risk of autoimmunity. However, based on the observation that SspE purified from the SspABCD-expressing *E. coli* cells

stays nuclease-active in in vitro conditions even in the absence of GTP, the possibility cannot be excluded that the open-to-closed transition does not occurs and therefore a single GTP hydrolysis enhancement is sufficient to activate SspE indefinitely.

In R-M defense systems, REases use sequence-specific nucleobase methylation to distinguish between host and invading foreign DNA and cleave unmodified invaders. Methyl groups are proposed to protect self-DNA from REases by preventing them from binding sites through steric hindrance[27]. This process is in line with the observation that the recognition sites in host DNA are nearly completely protected by methylation to avoid self-restriction. In contrast, only 21,788 out of 160,541 5′-CCA-3′ consensus sequences are PT modified in the *V. cyclitrophicus* FF75 genome, and PT distribution displays significant molecule-to-molecule heterogeneity even in the presence of active SspE[10,12]. Thus, two fundamental questions are raised concerning how SspE accomplishes self-nonself discrimination in the face of such a state of less-than-saturating PT modification and how the non-PT-protected regions in host DNA avoid self-restriction by SspE. Here, our data support the model that SspE is preferentially directed to PT sites, preventing the non-PT-modified regions in self-DNA from being targeted by SspE and subsequent autoimmunity. Importantly, SspE exhibits a slight difference in affinity for 5′-C$_{PS}$CA-3′-containing-DNA and unmodified DNA (K$_d$ values of 8.5 μM and 18.4 μM, respectively). Given the low frequency of PTs in bacterial genomes, the moderate PT affinity may be merely sufficient to prevent SspE from targeting 5′-CCA-3′ that are nearby PT sites in self-DNA rather than shifting SspE from foreign DNA. One can envisage a scenario where the majority of SspE is recruited to PT sites with strong binding affinity, preventing it from interacting with and attacking unmodified invasive DNA. Apparently, this possibility is inconsistent with the antiphage activity of SspE.

Notably, the constitutively active nuclease activity of SspE$_{K40A}$ may still be restrained by the PT modification. Due to the enhanced nuclease tolerance[28], the PT modification in self-DNA is assumed to tolerate the nuclease activity of SspE. However, PT-modified plasmid DNA isolated from SspABCD-expressing cells is still susceptible to SspE and its derivatives in vitro, similar to the unmodified counterparts. This discrepancy between the in vivo and in vitro observations suggests that additional elements in self-DNA are involved in preventing autoimmunity mediated by SspE, such as nuclease-resistant PT bonds, longer DNA recognition sequences and DNA secondary structures.

Collectively, SspE functions as a specific PT reader to sequentially modulate its GTPase activity, DNA binding affinity, and nickase activity, providing protection against phages and preventing autoimmunity, which highlights a coupled recognition-activating-nicking defense mechanism that differs from that of PT-based Dnd, methylation-based R-M, or other known defense mechanisms. SspABCD-SspE therefore represents a unique paradigm in bacterial antiphage defense mechanism that target nucleic acids.

## Methods

### Bacterial strains and plasmids
The bacterial strains and plasmids used in this study are listed in Supplementary Table 3.

### Protein expression and purification
The plasmids expressing SspE and its variants were transformed into chemically competent *E. coli* BL21(DE3) cells. Transformants were inoculated into 50 mL of Luria-Bertani (LB) broth and cultured at 37 °C overnight. The overnight culture was diluted 1:100 (v/v) into LB broth and grown to an OD$_{600}$ of 0.8 followed by protein induction using 0.2 mM isopropyl β-d-1-thiogalactopyranoside (IPTG) for 16-24 h. After cell collection by centrifugation, the cell pellet was resuspended in binding buffer (25 mM Tris-HCl, 300 mM NaCl, and 20 mM imidazole, pH 8.0) and disrupted using a cell homogenizer (JNBIO, Guangzhou, China). Cell debris was removed by centrifugation at 14,000 g for

45 min at 4 °C, and the supernatant was then purified by Ni$^{2+}$-NTA affinity chromatography (GE Healthcare) and Superdex 200 gel filtration chromatography (GE Healthcare). Peak fractions were combined and concentrated to 10 mg/mL for crystallization.

### Crystallization and structure determination
*S. scabiei* DSM 41658 SspE$_{CTD}$ crystals were grown by the hanging-drop vapor-diffusion method with buffer containing 14% PEG 4000, 0.1 M imidazole, and 0.2 M magnesium phosphate, pH 7.0 (Hampton Research, USA) at 14 °C. The crystals were stored in liquid nitrogen with cryoprotectant buffer containing 25% glycerol. Crystal diffraction data at a resolution of 2.7 Å were collected on the BL19U1 beamline at the National Center for Protein Sciences Shanghai (NCPSS, Shanghai, China) at 100 K and processed using HKL2000 software[29]. The PHASER[30,31] program was used to determine the SspE$_{CTD}$ crystal structure with the SAD method. Model building and refinement were performed using COOT[32] and REFMAC[13,30,33]. The crystals belonged to the C222$_1$ space group, and there were four molecules of SspE$_{CTD}$ in each asymmetric unit. The final refined model had an R$_{work}$/R$_{free}$ of 21.22%/26.50%.

Crystals of the full-length SspE from *S. yokosukanensis* DSM 40224 were obtained as previously described[8]. The structure was solved by the molecular replacement method using the structures of SspE$_{1-601aa}$ (PDB code: 6JIV [https://doi.org/10.2210/pdb6jiv/pdb]) from *S. yokosukanensis* DSM 40224 and SspE$_{CTD}$ from *S. scabiei* DSM 41658 as the search models. After refinement by the CCP4 program REFMAC[13,30,33], the final refined model of full-length SspE from *S. yokosukanensis* DSM 40224 had an R$_{work}$/R$_{free}$ of 20.1%/27.6%.

Crystals of SspE$_{R100A}$ from *S. yokosukanensis* DSM 40224 were grown in buffer containing 10% PEG 3350 and 4% Tacsimate, pH 7.0 (Hampton Research, USA) at 14 °C. Crystal diffraction data at a resolution of 3.48 Å were collected on the BL19U1 beamline at NCPSS (Shanghai, China) at 100 K and processed using HKL2000 software[29]. Through molecular replacement, the structure of SspE$_{R100A}$ was determined. The crystals belonged to the P2$_1$2$_1$2$_1$ space group, and there were four molecules of SspE$_{R100A}$ in each asymmetric unit. The final refined model had an R$_{work}$/R$_{free}$ of 25.7%/30.8%. The quality of the structure models was evaluated using the PROCHECK program[30], and the results indicated that the model exhibited good stereochemistry based on a Ramachandran plot.

### GTPase activity assay
The GTP hydrolysis activity of SspE and its mutants was assayed using the QuantiChrom ATPase/GTPase assay kit (BioAssay Systems) to measure the inorganic phosphate released through a chromogenic reaction with malachite green. In this assay, 1 μM protein, either alone or mixed with 1 μM 40-bp DNA fragment bearing PT-modified 5′-C$_{PS}$CA-3′ or nonmodified 5′-CCA-3′ motif, was incubated with 100 μM GTP in CutSmart Buffer (50 mM potassium acetate, 10 mM magnesium acetate, 100 μg/ml BSA, 20 mM Tris-acetate, pH 7.9, New England Biolabs, USA) at 28 °C for 30 min. Then, 200 μL of assay kit reagent was added, and the reaction was incubated for 30 min followed by a microplate reading at OD 620 nm. GTP hydrolysis in the absence of SspE was determined as background absorbance, and sample values were normalized by subtraction of the background.

### DNA nicking assay
The DNA nicking assay was performed as previously described by Xiong et al[8]. Briefly, 300 ng of pUC19 DNA was incubated with 2 μM SspE or its variants in CutSmart Buffer (New England Biolabs, USA) in a total volume of 10 μL at 28 °C for a time course or as indicated. Products were analyzed by electrophoresis on a 0.8% agarose gel.

### EMSA
Different concentrations of SspE and its mutants were incubated with 200 ng of linearized pUC19 in 15 μL of binding buffer (10 mM Tris-HCl,

100 mM NaCl, 0.01 mM EDTA, pH 8.0 and 5% glycerol) for 15 min at 30 °C. The reaction mixtures were electrophoresed on a 0.8% agarose gel at 80 V for 80 min. Data were quantified using Quantity One and plotted using GraphPad Prism software (version 6) as a best-fit curve through a nonlinear regression model using the following equation: $Y = B_{max} \times [C]/(K_d + [C])$, where $B_{max}$ is the maximum specific binding, [C] is the protein concentration, and Y is the ratio of specific binding.

### Complex structure prediction and interface contact residue computation

The complex formed by the interaction between DNA and protein was predicted by the docking method HoDock[22], which incorporates initial rigid docking and refined semiflexible docking. In total, 200,000 complex structures were generated and scored to obtain the final correct complex structure model. The docked model of the complex was also minimized using the molecular dynamics simulation package Gromacs 4.5[34]. The interface contact residues were calculated by the Cartesian coordinates of nonhydrogen heavy atoms C, N, and O. Residues with at least one pair of heavy atoms within 4 Å were denoted interface contact residues.

### NM analysis

The structural coordinates of SspE were submitted for the NM analysis using the Elastic Network Model server (http://www.sciences.univ-nantes.fr/elnemo/index.html)[35], a fast and simple tool to compute the low-frequency normal modes of a protein. The following parameters were used for the calculation: NMODES = 5, DQMIN = − 100, DQMAX = 100 and DQSTEP = 20. The major vibrational modes generated by the server were used for further analysis.

In the output of the normal mode analysis server, the calculated normal modes are listed according to their importance (the mode slower in vibrational frequency is considered more important). The 7th normal mode in this list, which is the 1st vibrational mode (the first six normal modes are translational and rotational modes of the system as a whole), is the one with the slowest vibrational frequency and thus is regarded as the most important vibrational mode. As the total numbers of vibrational modes of our systems were too high, we selected only the major mode (the 7th normal mode of SspE) for further analysis.

### FRET analysis

A 717-bp fragment encoding cyan fluorescent protein (CFP) was amplified from pET28a-CFP plasmid DNA using primer #1 (CTGAGA ATTCCATCACCATGTGAGCAAGGGCGAG) and primer #2 (AATGCGG CCGCTTACTTGTACAGCTCGTC). The fragment was digested with EcoRI and NotI and inserted into pET28a, which had been digested by the same enzymes to generate pPT562. The fragment encoding YFP was amplified from pET28a-YFP plasmid DNA using primer #3 (CGCCATA TGGTGAGCAAGGGCGAGGAG) and primer #4 (GTGATGGTGGTGATGA TGCTTGTACAGCTCGTCCAT). A 2,313-bp sequence encoding SspE was amplified from the genomic DNA of *S. yokosukanensis* DSM 40224 using primer #5 (CATCATCACCACCATCACATGGAGACTAAAGAGATC) and primer #6 (AGAATGAATTCCGGCTCGAATCCCAGCC). The two PCR products harboring the *yfp* and *sspE* genes were fused using primer #3 and primer #6 using overlap extension PCR. The resulting YFP-sspE fragment was digested with NdeI and EcoRI and ligated into pPT562, which had been digested by the same enzymes, generating pPT563. A construct of pPT564 expressing both YFP and CFP was also generated as a control. All the final constructs were confirmed by sequencing and expression in *E. coli* BL21(DE3).

FRET measurements were performed using a Cytation™3 spectrofluorometer that allowed simultaneous monitoring of CFP and YFP fluorescence emissions. The spectrofluorometer was equipped with a fluorescence monochromator offering a variable bandwidth from

250 nm to 700 nm and a filter with a variable bandwidth from 200 nm to 700 nm to separate the CFP from the YFP signal. LEDs were used as light sources, and a Sony CCD camera (1.1 million pixels) was used as the detector in the imaging system. Purified fluorescent proteins were assayed at a final concentration of 0.2 μM in buffer (10 mM Tris-HCl, 100 mM NaCl, 1 mM Mg²⁺, pH 8.0). The emission spectra of CFP and YFP were obtained by excitation at 430 nm and detected simultaneously through a 450 to 600-nm bandpass filter. Moreover, the emission fluorescence of YFP-SspE at 530 nm after excitation at 430 nm was determined and subtracted from the fluorescent signals measured from the YFP-SspE-CFP construct. FRET was expressed as the ratio of the maximum fluorescence intensity of the spectral peaks corresponding to YFP and CFP[36].

### Microscale thermophoresis (MST)

A monolith instrument (NT.115) (NanoTemper)[37] was used to measure the interaction between the protein and DNA fragments labeled with Alexa488. The 5′-C$_{PS}$CA-3′-containing DNA fragment was PCR-amplified from pUC19 using primer #7 (Alexa488-CTGAGAGTGCAC$_{PS}$CATATGC G) and primer #8 (TCCTGTTTTTGCTCACC$_{PS}$CAG). Primers #9 (Alexa 488-CTGAGAGTGCACCATATGCG) and #10 (TCCTGTTTTTGCTCACC CAG) were used to produce 5′-CCA-3′-containing DNA. The concentration of Alexa Fluor 488-labeled DNA was kept constant (12.5 nM), while the concentration of SspE and its variants were varied. The samples were incubated at room temperature for 15 min in MST buffer (10 mM Tris-HCl, 100 mM NaCl, pH 8.0). Next, the samples were loaded into MST capillaries, and analyses were performed.

### Molecular graphics

All protein structure figures were generated using the PyMOL program (http://www.pymol.org). Sequence conservation of CTD mapped onto the surface of its crystal structure was generated by the ConSurf server (http://consurf.tau.ac.il)[38].

### Reporting summary

Further information on research design is available in the Nature Portfolio Reporting Summary linked to this article.

## Data availability

The data supporting the findings of this study are available from the corresponding author on request. The coordinates and structure factors of SspE$_{CTD}$ from *S. scabiei* DSM 41658 and full-length SspE and SspE$_{R100A}$ from *S. yokosukanensis* DSM 40224 generated in this study have been deposited in the Protein Data Bank under accession numbers 7DRI [https://doi.org/10.2210/pdb7dri/pdb], 7DRS [https://doi.org/10.2210/pdb7drs/pdb], and 7DRR [https://doi.org/10.2210/pdb7drr/pdb], respectively. The source data for Figs. 1 to 5, Supplementary Figs 1 to 4 are provided as a Source Data file. Source data are provided with this paper.

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

## Acknowledgements

We thank the staff from the BL19U1 beamline of the National Facility for Protein Science in Shanghai (NFPS) at Shanghai Synchrotron Radiation Facility for assistance during the data collection. This work was supported by grants from the National Key Research and Development Program of China (2022YFA0912500 to S.C., 2022YFA0912200 to L.W., and 2020YFA0907300 to G.W.), the National Natural Science Foundation of China (31925002 to S.C., 32125001 and 31720103906 to L.W., and 31872627 to G.W.), and Shanghai Jiao Tong University Scientific and Technological Innovation Fund (to G.W.).

## Author contributions

G.W., S.C. and L.W. conceived the study and supervised the project. H.G., X.G., J.Z., Y.Z., and Y.W. performed the experiments. H.G., X.G., J.Z., Y.Z., J.D., Y.W., L.C., Z.D., J.W., S.C., G.W., and L.W. analyzed the data. H.G., S.C., G.W., and L.W. wrote the manuscript.

## Competing interests

The authors declare no competing interests.
