## [Peer Review File · Nature Communications]

Nicking mechanism underlying the DNA
phosphorothioate-sensing antiphage defense by SspE.Editorial Note: This manuscript has been previously reviewed at another journal that is not operating a transparent peer review scheme. This document only contains reviewer comments and rebuttal letters for versions considered at *Nature Communications*.

REVIEWERS' COMMENTS

Reviewer #1 (Remarks to the Author):

The authors submit a revised manuscript that satisfactorily addresses my comments and questions on the earlier version(s) of the manuscript. The authors provide a working model of the SspE effector of the SspABCD phosphothioate-sensing antiphage system. Several open questions concerning this model are indicated in the manuscript and data overinterpretation was reduced.

Reviewer #2 (Remarks to the Author):

The manuscript from Gao et al describes the activity of PT-sensing SspE, as part of phage defence.

This is the third time this reviewer has read the manuscript.

I'm pleased to say the authors have worked hard and convinced me, well done.

By providing more data, and also reducing the scope of their model, the manuscript is tighter, easier to follow, and all claims are justified.

It will be interesting to see in due course how "open" conformation activity selects phage DNA over self, and how it closes again! Once it binds to non-PT DNA and nicks, if there is no PT, is it trapped in some way? However - these are questions for future work.

I have some minor suggestions:

Line 185 - I do not see the extra motifs in 2b ie 5'-GpsAAC and 5'-GpsGCC. Am I missing something?

Line 203 - I think you mean 2b

Line 226 - Please introduce the section with motivation for doing EMSAs, as it dives in

Line 266 - Again, please add a sentence to explain why you made the linker mutant

Line 318 - change to "an" not "a"

Reviewer #3 (Remarks to the Author):

Gao, Gong, et al., report in the submitted manuscript how SspE, a DNA nickase, senses phosphorothioate modifications in the host genome and confers phage resistance in bacteria. This is a solid study that provides new insight into the function of the unusual epigenetic mark. I support publication of this paper in Nature Communications. The following suggestions are for the authors to consider.

Phosphorothioate modifications may present a new restriction modification system not yet fully appreciated by the microbiology field. Unlike other well-characterized restriction modification systems, such as adenine methylation, the PT-sensing and cleavage mechanism seems a lot more complicated. The complexity of the system may go beyond the scope of one manuscript. I don't think the value of the study will be tarnished by a model that still has question marks. In contrast, overinterpretation of data and premature conclusions may offset future research efforts. I suggest the authors to reconsider their model in the following aspects:

1, the authors mentioned a couple of times that SspE was expressed in PT-expressing cells and was therefore capable of nicking DNA in the absence of GTP. But they also suggest in line 312: "This finding confirmed that GTP hydrolysis led to a longer distance between the NTD and CTD, in agreement with the conformational transition from the 'closed' to the 'open' form". In fact, throughout the study, the authors indicate sequential events of PT-DNA loading, GTP hydrolysis, conformational change of SspE (and hence its DNA binding affinity and nickase activity). Protein expression in PT-expressing cells only

means that SspE may have encountered PT-modified DNA before (I assume PT-DNA was not co-purified, since the affinity is low and copurification of DNA is easy to verify). In in vitro conditions without GTP, I find it hard to imagine that the protein will stay active permanently, unless the authors are suggesting hydrolysis of a single GTP molecule is sufficient to activate SspE indefinitely. Or GTP hydrolysis and DNA nicking are two independent events, both of which are consequences of loading of PT-DNA. Either way, the authors should clarify.

2, the authors suggest SspE-K40A is still sensing PT-DNA as “the SspABCD-SspEK40A module still provided protection against phage infection at the same level as wild-type SspABCD-SspE (Fig. 2c), confirming that the PT responsiveness did not disturb the restriction of foreign DNA while avoiding self-targeting (line 252)”. However, Fig. 1c, 2b, and 4f do not support the claim. The presence of the putative PT-binding site does not translate into intact PT sensing, not to mention that the sequence specificity of PT sensing was never explained, which adds a whole new level of complexity to the current model.

3, the quality of the microscale thermophoresis data is imperfect. Line 333: this preferential affinity to PT-DNA was not detected for the SspE Y30A and SspEQ31A mutants, which showed abrogated PT responsiveness. Are the authors suggesting the apparent affinity is a sum of NTD (PT responsive) and CTD (PT irresponsive). If so, they should point it out. Can the authors provide a model in which SspE binds to the same piece of DNA with both binding sites?

4, the original line 418 “no apparent anti-plasmid activity of SspE in E. coli was observed” should not be deleted. The discrepancy between in vitro and in vivo studies should not be ignored.

Reviewer #1 (Remarks to the Author):

The authors submit a revised manuscript that satisfactorily addresses my comments and questions on the earlier version(s) of the manuscript. The authors provide a working model of the SspE effector of the SspABCD phosphothioate-sensing antiphage system. Several open questions concerning this model are indicated in the manuscript and data overinterpretation was reduced.

Answer: We are so glad to hear that reviewer #1 are fully satisfied with our responses.

Reviewer #2 (Remarks to the Author):

The manuscript from Gao et al describes the activity of PT-sensing SspE, as part of phage defence.

This is the third time this reviewer has read the manuscript.

I'm pleased to say the authors have worked hard and convinced me, well done.

By providing more data, and also reducing the scope of their model, the manuscript is tighter, easier to follow, and all claims are justified.

It will be interesting to see in due course how "open" conformation activity selects phage DNA over self, and how it closes again! Once it binds to non-PT DNA and nicks, if there is no PT, is it trapped in some way? However - these are questions for future work.

Answer: Indeed, it would be interesting to see in due course how SspE switches from the "closed" to "open" conformation and vice versa to accomplish the self/nonself discrimination and defense. It will be the focus of our future work.

I have some minor suggestions:

Line 185 - I do not see the extra motifs in 2b ie 5'-GpsAAC and 5'-GpsGCC. Am I missing something?

Answer: The influence of motifs, including 5'-GpsAAC-3' and 5'-GpsGCC-3',

on the GTPase activity of SspE *in vitro* had been described in our previous work (ref. 6). The reference "6" was displayed after the 5'-G_{PS}GCC-3'. Fig. 2 shows the effect of these motifs on the GTPase activity of SspE when they occur in a single-stranded DNA or RNA molecule.

Line 203 - I think you mean 2b.

Answer: Revised as suggested.

Line 226 - Please introduce the section with motivation for doing EMSAs, as it dives in

Answer: The sentence has been rephrased for clarification.

Line 266 - Again, please add a sentence to explain why you made the linker mutant

Answer: For clarification, a sentence has been added to explain why we made the linker mutant.

Line 318 - change to "an" not "a"

Answer: Thank you. Revised as suggested.

Reviewer #3 (Remarks to the Author):

Gao, Gong, et al., report in the submitted manuscript how SspE, a DNA nickase, senses phosphorothioate modifications in the host genome and confers phage resistance in bacteria. This is a solid study that provides new insight into the function of the unusual epigenetic mark. I support publication of this paper in Nature Communications. The following suggestions are for the authors to consider.

Phosphorothioate modifications may present a new restriction modification system not yet fully appreciated by the microbiology field. Unlike other well-characterized restriction modification systems, such as adenine methylation, the PT-sensing and cleavage mechanism seems a lot more complicated. The complexity of the system may go beyond the scope of one manuscript. I don't think the value of the study will be tarnished by a model that still has question marks. In contrast, overinterpretation of data and premature conclusions may offset future research efforts. I suggest the authors to reconsider their model in the following aspects:

1, the authors mentioned a couple of times that SspE was expressed in PT-expressing cells and was therefore capable of nicking DNA in the absence of GTP. But they also suggest in line 312: “This finding confirmed that GTP hydrolysis led to a longer distance between the NTD and CTD, in agreement with the conformational transition from the ‘closed’ to the ‘open’ form”. In fact, throughout the study, the authors indicate sequential events of PT-DNA loading, GTP hydrolysis, conformational change of SspE (and hence its DNA binding affinity and nickase activity). Protein expression in PT-expressing cells only means that SspE may have encountered PT-modified DNA before (I assume PT-DNA was not co-purified, since the affinity is low and copurification of DNA is easy to verify). In *in vitro* conditions without GTP, I find it hard to imagine that the protein will stay active permanently, unless the authors are suggesting hydrolysis of a single GTP molecule is sufficient to activate SspE indefinitely. Or GTP hydrolysis and DNA nicking are two independent events, both of which are consequences of loading of PT-DNA. Either way, the authors should clarify.

Answer: Thank you for pointing out the two ways regarding the difference between *in vivo* and *in vitro* nickase results of SspE. The current results left unanswered the question of whether SspE undergoes an opposite open-to-closed conformational change once its GTPase activity is no longer sufficient to maintain the open state of SspE. After all, the constitutively active nickase activity of SspE in bacterial cells is predicted to increase the risk of autoimmunity. However, based on the observation that SspE purified from the SspABCD-expressing *E. coli* cells stays nuclease-active in *in vitro* conditions even in the absence of GTP, the possibility cannot be excluded that the open-to-closed transition does not occur and therefore a single GTP hydrolysis enhancement is sufficient to activate SspE indefinitely. This information has been added in the Discussion section. As to the other possibility that the N-terminal GTP hydrolysis and C-terminal DNA nicking are two independent consequences of loading of PT-DNA, our results do not support this. The replacement of R100, a key residue for GTP hydrolysis, by alanine simultaneously abolished the GTPase activity and nickase activity of SspE (PMID: 32251370, Figure 4c and 4d). In addition, the constitutively stimulated GTPase activity leads to enhanced nickase activity of SspE_{K40A} (Figure 1C).

2, the authors suggest SspE-K40A is still sensing PT-DNA as “the SspABCD-SspEK40A module still provided protection against phage infection at the same level as wild-type SspABCD-SspE (Fig. 2c), confirming that the PT

responsiveness did not disturb the restriction of foreign DNA while avoiding self-targeting (line 252)". However, Fig. 1c, 2b, and 4f do not support the claim. The presence of the putative PT-binding site does not translate into intact PT sensing, not to mention that the sequence specificity of PT sensing was never explained, which adds a whole new level of complexity to the current model.

Answer: Thank you for pointing this out. Line 252 should be line 352. The sentence has been rephrased as: The SspABCD-SspE_{K40A} module still provided protection against phage infection at the same level as wild-type SspABCD-SspE (Fig. 2c), highlighting the essential role of the hydrophobic patch-mediated PT sensing rather than PT-binding in preventing autoimmunity.

3, the quality of the microscale thermophoresis data is imperfect. Line 333: this preferential affinity to PT-DNA was not detected for the SspE Y30A and SspEQ31A mutants, which showed abrogated PT responsiveness. Are the authors suggesting the apparent affinity is a sum of NTD (PT responsive) and CTD (PT irresponsive). If so, they should point it out. Can the authors provide a model in which SspE binds to the same piece of DNA with both binding sites?

Answer: Indeed, the PT binding preference of SspE is mediated by the sum of NTD and CTD. The sentence has been rephrased to point it out according to the reviewer's suggestion.

As we can see in Figure 5 and Supplementary Fig. 2, no direct DNA binding between the NTD and DNA was detected by EMSAs. PT is involved in directing SspE to PT sites by virtue of the hydrophobic patch in the NTD and the DNA binding affinity is mediated by the CTD. Thus, SspE binds to a piece of DNA with only CTD instead of both binding sites at the same time. The N-terminal PT-sensing hydrophobic patch is responsible for directing SspE to preferentially bind PT sites.

4, the original line 418 "no apparent anti-plasmid activity of SspE in E. coli was observed" should not be deleted. The discrepancy between in vitro and in vivo studies should not be ignored.

Answer: As suggested, the sentence has been added back in the text and Supplementary Figure 4.